# Association between Circulating Amino Acids and COVID-19 Severity

**DOI:** 10.3390/metabo13020201

**Published:** 2023-01-29

**Authors:** Ina Maltais-Payette, Fannie Lajeunesse-Trempe, Philippe Pibarot, Laurent Biertho, André Tchernof

**Affiliations:** 1Quebec Heart and Lung Institute, Quebec City, QC G1V 4G5, Canada; 2School of Nutrition, Faculty of Agriculture and Food Sciences, Laval University, Quebec City, QC G1V 0A6, Canada; 3Department of Medicine, Faculty of Medicine, Laval University, Quebec City, QC G1V 0A6, Canada; 4Department of Surgery, Faculty of Medicine, Laval University, Quebec City, QC G1V 0A6, Canada

**Keywords:** BQC19, COVID-19, amino acids, obesity

## Abstract

The severity of the symptoms associated with COVID-19 is highly variable, and has been associated with circulating amino acids as a group of analytes in metabolomic studies. However, for each individual amino acid, there are discordant results among studies. The aims of the present study were: (i) to investigate the association between COVID-19-symptom severity and circulating amino-acid concentrations; and (ii) to assess the ability of circulating amino-acid levels to predict adverse outcomes (intensive-care-unit admission or hospital death). We studied a sample of 736 participants from the Biobanque Québécoise COVID-19. All participants tested positive for COVID-19, and the severity of symptoms was determined using the World-Health-Organization criteria. Circulating amino acids were measured by HPLC-MS/MS. We used logistic models to assess the association between circulating amino acids concentrations and the odds of presenting mild vs. severe or mild vs. moderate symptoms, as well as their accuracy in predicting adverse outcomes. Patients with severe COVID-19 symptoms were older on average, and they had a higher prevalence of obesity and type 2 diabetes. Out of 20 amino acids tested, 16 were significantly associated with disease severity, with phenylalanine (positively) and cysteine (inversely) showing the strongest associations. These associations remained significant after adjustment for age, sex and body mass index. Phenylalanine had a fair ability to predict the occurrence of adverse outcomes, similar to traditionally measured laboratory variables. A multivariate model including both circulating amino acids and clinical variables had a 90% accuracy at predicting adverse outcomes in this sample. In conclusion, patients presenting severe COVID-19 symptoms have an altered amino-acid profile, compared to those with mild or moderate symptoms.

## 1. Introduction

The symptoms associated with coronavirus disease 2019 (COVID-19) are very heterogenous. Around 80% of people infected are asymptomatic or present mild symptoms and do not require hospitalization [1]. This leaves a significant percentage (20%) of patients needing medical care, a quarter of whom require extensive treatments such as assisted ventilation [1]. In order to provide the best care for patients and to organize health care efficiently, it is important to understand the pathophysiology of the disease and to find relevant predictive markers of severe symptoms.

Since the beginning of the pandemic, an impressive number of studies investigating metabolomics in relation to COVID-19 have been published. Most of the studies aimed to either define the unique metabolic signature of COVID-19 or to predict the severity of and/or mortality from the disease. Reviews of this literature are now available [2,3,4] and they highlight the fact that one of the groups of metabolites most often altered in the context of severe COVID-19 symptoms is circulating amino acids. However, the results for individual amino acids are sometimes discordant. For example, four studies have reported that phenylalanine level was positively associated with symptom severity [5,6,7,8], whereas one study reported an inverse association [9], and four studies reported no association [10,11,12,13]. Moreover, very few studies have evaluated the potential of using circulating amino acids to identify patients who will develop life-threatening symptoms.

Studies have demonstrated that patients living with obesity are at greater risk of presenting severe symptoms when infected with SARS-CoV-2, compared to those without obesity [14]. Moreover, visceral adiposity appears to be a better predictor of disease severity compared to subcutaneous adiposity [15]. It has been suggested that the low-grade chronic inflammation associated with visceral obesity could exacerbate the inflammation storm observed in COVID-19 [14]. Circulating levels of some amino acids have been shown to be positively associated with overall adiposity, the most studied being branched-chain amino acids (BCAAs, namely leucine, isoleucine and valine) [16]. We have previously demonstrated that circulating glutamate level was significantly associated with waist circumference and visceral adiposity, more strongly than any other amino acid [17,18,19]. The aims of the present study were: i) to determine which circulating amino acids are associated with COVID-19-symptom severity; and ii) to determine whether they could be relevant predictors of adverse outcomes. We hypothesized that amino acids previously shown to be associated with adiposity indices (leucine, isoleucine, valine and glutamate) were the most strongly associated with symptom severity and the best predictors of adverse outcomes. This study is targeted on amino acids and does not aim to find the metabolite (or group of metabolites) most strongly associated with COVID-19 severity.

## 2. Materials and Methods

### 2.1. Study Population and End-Points

We studied a sample of the Biobanque Québécoise COVID-19 (BQC19) [20], a biobanking infrastructure in the province of Quebec that aims to provide researchers with data and biological samples to perform studies on COVID-19. It was initiated in the spring of 2020, and recruitment of patients took place in 11 different hospitals across the province. The BQC19 inclusion criteria were: i) having a polymerase-chain-reaction (PCR)-diagnostic test for COVID-19 (be it positive or negative); and ii) being able to provide informed consent. The BQC19 is an ongoing organization, and the last update available for analyses spans from February 2020 to August 2022. The following variants were present in the province of Quebec during this period: delta (B.1.617.2), omicron (B.1.1.529), alpha (B.1.1.7), beta (B.1.351) and gamma (P.1) [21].

COVID-19 severity was determined at inclusion based on the World Health Organization (WHO) Clinical Progression Scale as mild, moderate, severe or deceased [22]. There were very few deceased patients at inclusion; therefore, we combined the severe and deceased categories for all analyses (subsequently referred to as “severe”).

For the present study, we excluded: (i) patients without metabolomic measurement from blood taken within 7 days of inclusion; (ii) patients who tested negative for COVID-19; (iii) patients without a disease-severity score at inclusion; and (iv) pregnant women and children. This resulted in a sample of 736 individuals. A flowchart of patient selection is presented in Figure 1.

To determine the ability of amino acids to predict adverse outcomes, we considered a subsample of 476 hospitalized patients with data on ICU admission and/or vital status at discharge, and for whom metabolomic measurements were carried out using plasma collected before their ICU admission or death. We defined adverse outcomes as admission to the intensive care unit (ICU) or hospital death.

### 2.2. Metabolomics Measurements

Plasma samples were analyzed by Metabolon Inc. [23]. Briefly, samples were mixed with methanol, centrifuged and analyzed using ultra-performance liquid chromatography and tandem mass spectrometry (UPLC-MS/MS) using both reverse-phase and hydrophilic-interaction liquid chromatography and both positive and negative ionization. Pooled samples and blanks were used to measure extraction and injection variability. Overall, 1438 metabolites were measured, including 1155 identified compounds and 280 unknown compounds. For more details about the protocol, see [23].

We focus on circulating amino acids for two reasons. First, abdominal obesity has previously been associated with COVID-19 severity, and we have previously demonstrated that circulating concentrations of some amino acids are correlated with central fat accumulation. Second, the existing literature shows that amino acids, as a group, are almost invariably altered in the context of severe COVID-19, although results for specific amino acids are heterogeneous. When more than one metabolomic measurement was available in our 7-day timeframe, we used the closest to study inclusion.

### 2.3. Statistical Analyses

We compared patient characteristics and comorbidities between severity groups, using Krustal–Wallis and chi-squared tests. To examine the association between disease severity and circulating-amino-acid levels, we used univariate and adjusted logistic-regression models comparing mild patients to moderate and severe patients. We reported the results as odd ratios (OR) for each 1 SD increment in circulating amino acids. We compared the level of metabolites annotated to the TCA and urea cycle between severity groups using the Krustal–Wallis test.

To determine the ability of individual amino-acid levels to predict adverse outcomes, we performed univariate logistic-regression models and computed receiving-operator-characteristic (ROC) curves. The latter is a graphical tool that allows the visualization of the trade-off between sensitivity and specificity of a given predictor. It also allows the calculation of the area under the ROC curve (ROC_AUC), which is a metric of the ability of the predictor to discriminate between two groups. For example, a ROC_AUC of 0.70 indicates that the predictor has 70% accuracy at classifying observations in the correct group. A predictor is considered good when it has a ROC_AUC above 0.8, and excellent above 0.9 [24]. We did not identify optimal thresholds from the ROC curves because the amino-acid levels were expressed as relative abundance, and therefore thresholds would have been of limited use.

We also ran univariate logistic analysis on clinical variables for comparison purposes. Additionally, we built a multivariate logistic-regression model using circulating-amino-acid levels and the clinical variables significantly associated with the outcome in univariate analysis as candidate predictors. Missing values were imputed using multivariate imputation by chained equations using predictive mean matching for continuous variables and logistic regression for ordinal variables. Predictive variables were scaled, and those with ≥50% missing values were discarded. Fifty imputations were run, and forward as well as backward logistic-models were fit with the candidate variables on each imputation. The variables used in the majority (≥50%) of models were selected for the final model. The final model was run on the imputed data, and the results were pooled.

A *p*-value below 0.05 was considered significant, except for post hoc analyses, where the Bonferroni correction was applied.

## 3. Results

Overall, 736 individuals were included for analysis (Figure 1). According to WHO criterion for COVID-19 disease severity [22], 159 had mild symptoms, 353 moderate symptoms and 224 severe symptoms.

Sample characteristics by severity are presented in Table 1. Patients with moderate and severe symptoms were significantly older than those with mild symptoms. Sex was also associated with severity, with the severe subgroup counting more males than the other two groups. Glucose level was significantly higher in patients with severe symptoms. C-reactive protein (CRP), creatinine, urea, and white-blood-cell count were significantly higher, whereas albumin and hemoglobin were lower in severe vs. mild and moderate patients. Although the percentage of participants vaccinated was not different across groups, the number of doses received differed significantly. Indeed, there was a greater prevalence of participants with three vaccine doses in the mild group (19%) compared to the moderate and severe groups (1%).

Among comorbidities (see Appendix A), the prevalence of diabetes, dementia, obesity, arterial hypertension, atrial fibrillation, chronic kidney disease, chronic neurological disease, strokes, hematologic disease, coronary artery disease, rheumatologic disease, cancers and heart failure was significantly associated with COVID-19-symptom severity. However, once adjusted for age, the association was no longer significant for dementia, hematologic disease, rheumatologic disease, cancers and heart failure (results not shown).

Among the participants living with obesity (*n* = 179 overall), 118 (66%) also had one or more metabolic comorbidities (diabetes, arterial hypertension, or coronary arterial disease). The prevalence of individuals with obesity and a metabolic comorbidity was significantly higher in the moderate and severe COVID-19-symptom severity groups, compared to the mild group (chi-square *p*-value 0.0084).

The number of participants taking angiotensin-converting enzyme (ACE) inhibitors, corticosteroids, anticoagulants and oral hypoglycemic agents or insulin at inclusion was significantly higher in the moderate and severe groups, compared to the mild group (Appendix A).

### 3.1. Are Circulating Amino Acids Associated with COVID-19 Severity?

The mean number of days between study inclusion (and therefore disease-severity assessment) and blood draw for metabolomics measurement was 4 days. We first evaluated the association between circulating-amino-acid concentrations and COVID-19 severity, using univariate logistic-models. The results from this analysis are presented as the odds of having mild vs. severe or mild vs. moderate symptoms for each increment of 1 SD in the circulating amino acid (Table 2).

Out of the 20 amino acids tested, 16 were significantly associated with COVID-19 severity. In this univariate analysis, the amino acids most strongly associated with severe symptoms were phenylalanine (positively associated) and cysteine (inversely associated). Concentrations of the 3 BCAAs and glutamate were also significantly associated with severity, but to a lesser extent.

To evaluate the impact of potentially cofounding variables on the association between amino acids and disease severity, we adjusted the logistic models for characteristics found to be significantly different between severity groups. The results of this analysis are presented in Figure 2.

Adjustments for albumin levels had the greatest impact on the association between circulating amino acids and COVID-19 severity; even associations with the most significant amino-acid levels were no longer significant, once adjusted for this covariate. Overall, age, sex and body mass index (BMI) had little impact on the association between amino-acid concentrations and COVID-19-symptom severity.

We also ran models adjusted for comorbidities and medication intake at inclusion (see Appendix A). Overall, adjustment for comorbidities had very little effect on the associations. Medication intake also had little effect, except for oral hypoglycemic agents or insulin, which tended to increase the significance of some associations.

### 3.2. Are Circulating-Amino-Acid Levels Predictive of Adverse Outcomes?

Among the 476 participants with data on ICU admission and/or vital status at discharge, 26 were admitted to the ICU and 76 died in the hospital, giving a total of 102 patients with adverse outcomes. Blood was drawn on average 19 days before death (range: 1 to 111 days) and 3 days before ICU admission (range: 1 to 21 days).

The characteristics, comorbidities, medication at inclusion and medication during hospitalization, including treatment used for COVID-19, are presented in Appendix A. There was a greater proportion of men among patients who suffered from an adverse outcome. CRP, creatinine, lactate dehydrogenase and procalcitonin measured at study inclusion were higher in the adverse-outcome group. There was a greater prevalence of atrial fibrillation, arterial hypertension, myocardial infraction, coronary artery disease and chronic kidney disease among the adverse-outcome group.

We evaluated the ability of amino acids to distinguish participants who would suffer from adverse outcomes from those who would not, using logistic-regression models and reporting the predictive ability as ROC_AUC. To compare the performance of amino acids to that of variables often measured in a clinical context, we also ran this analysis for clinical variables significantly different across disease-severity categories. Results are presented in Figure 3for amino acids and in Appendix A for clinical variables. Circulating-amino-acid levels had a poor-to-average predictive ability, with ROC_AUC ranging from 0.49 (*p* = 0.6001) for tyrosine to 0.72 (*p* = 3.5 × 10^−11^) for phenylalanine. Clinical variables had similar or better predictive-abilities, with ROC_AUC ranging from 0.51 for ALT (*p* = 0.1763) to 0.82 for procalcitonin (*p* = 0.0019).

We used the circulating-amino-acid levels and clinical variables significantly associated with adverse outcome in the univariate models as candidate variables for the multivariate model. After removal of variables with ≥50% missing values, scaling and imputation, we used backward and forward stepwise-models to identify 10 variables to include in the final model: CRP, urea, arterial oxygen saturation (SaO_2_), SBP, phenylalanine, tryptophan, arginine, lysine, glutamate and serine. The final model is presented in Table 3. It had a ROC_AUC of 0.90 (95%CI = 0.86–0.93) for predicting adverse outcomes.

## 4. Discussion

In this analysis, we aimed to determine whether circulating-amino-acid concentrations were associated with COVID-19-symptom severity, and whether they could predict adverse outcomes in hospitalized patients. We found that 16 out of 20 amino acids were different between severe and mild COVID-19 symptoms. The strongest positive association was observed for phenylalanine, and the strongest inverse association was observed for cysteine. Phenylalanine also had a fair ability to predict adverse outcomes. To our knowledge, this is the largest study investigating circulating amino acids in the context of COVID-19.

A summary of the existing literature on circulating amino acids and COVID-19 severity is provided in Appendix A. Our results are concordant with previous reports of a significant association between severe COVID-19 and circulating levels of leucine, isoleucine, valine, glutamate and phenylalanine (positively) as well as glutamine, tryptophan, histidine, alanine, proline and cysteine (inversely) (Appendix A). We found a larger number of amino acids to be significantly associated with severity than previous studies, which could be explained by our larger sample size. We also confirmed that age, male sex and obesity are significant risk factors for developing severe symptoms when infected with SARS-CoV-2 [25].

Considering that obesity is associated with greater risk of suffering from severe COVID-19 symptoms, we hypothesized that circulating amino acids previously linked to obesity would be the strongest correlates of symptom severity and the best at predicting adverse outcomes. However, this hypothesis did not hold true, as the strongest associations with symptom severity were found for phenylalanine and cysteine, which are not systematically associated with obesity. Moreover, our results showed that the associations between amino-acid levels and disease severity remained significant after adjusting for BMI. Finally, although we would expect amino acids previously reported to be positively associated with adiposity to be also positively associated with COVID-19 severity, this was not always the case. For example, tryptophan is generally positively associated with measurements of adiposity [26], but it was inversely associated with symptom severity in this study. Overall, these results indicate that the state of severe COVID-19 affects amino-acid metabolism beyond what can be observed with elevated adiposity.

We did not find circulating-amino-acid concentrations to be good predictors of ICU admission and hospital death. Indeed, other lab measurements that have been validated in the literature and are much more accessible had similar or better predicting-abilities [27,28,29]. For example, procalcitonin had an accuracy of 82%, whereas the best performing amino acid, phenylalanine, had an accuracy of 72% in identifying patients who would suffer from an adverse outcome. Our multivariate logistic model that included both circulating amino acids and clinical variables had a 90% accuracy for predicting adverse outcomes. This indicates that an index including many relevant variables may perform better than a single biomarker. However, we consider this analysis preliminary, since the model was not validated on an independent cohort and because of the variability of elapsed time between blood draw and adverse-outcome occurrence (from 1 to 111 days). More studies are needed to determine the added value of amino acids to clinical variables for adverse-outcome prediction.

The strengths of this study include the rather large sample-size and the use of the WHO symptom-severity scale. Limitations include the absence of an uninfected control group and the large number of missing data for some variables. Other limitations include the lack of measurement of body-fat distribution and the variability in elapsed time between blood draw and adverse-outcome occurrence. Finally, we acknowledge that amino acids are more easily and frequently measured in metabolomics studies compared to other classes of metabolites, and this has probably contributed to the abundance of studies linking amino acids and COVID-19. Whether amino acids are better than other metabolites at predicting COVID-19 severity or adverse outcomes should be investigated further.

In conclusion, severe COVID-19 symptoms are associated with altered levels of many circulating amino acids, possibly hinting at global changes in nitrogen metabolism. These associations are mostly independent of age, sex and BMI.

## Figures and Tables

**Figure 1 metabolites-13-00201-f001:**
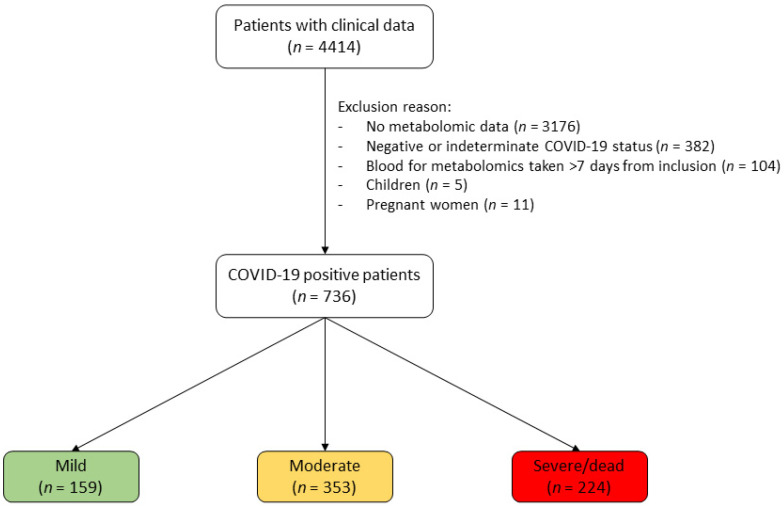
Patient-selection flowchart. PCR tests were used to assess COVID-19 status. Symptom severity was established at inclusion, using the World Health Organization (WHO) criteria [22].

**Figure 2 metabolites-13-00201-f002:**
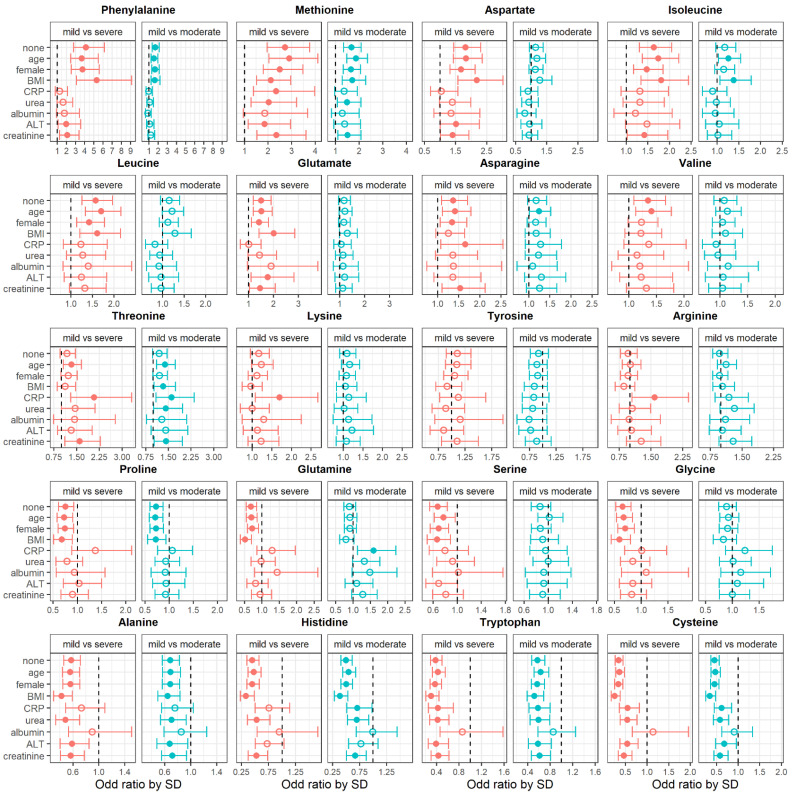
Odds of having mild vs. severe, or mild vs. moderate COVID-19 symptoms associated with circulating-amino-acid levels unadjusted (none) and adjusted for clinical variables. Results are presented as odds ratios per 1 standard-deviation (SD) increase in amino-acid level. The number of observations for each models varies; see Table 1 for the number of observations available for each variable. The *x*-axis scale is different for each amino acid. Full circles indicate a significant association and empty circles indicate a non-significant association. BMI: body mass index, CRP: C-reactive protein, ALT: alanine aminotransferase. For CRP, patients receiving tomizumab were excluded because it negates the measurement.

**Figure 3 metabolites-13-00201-f003:**
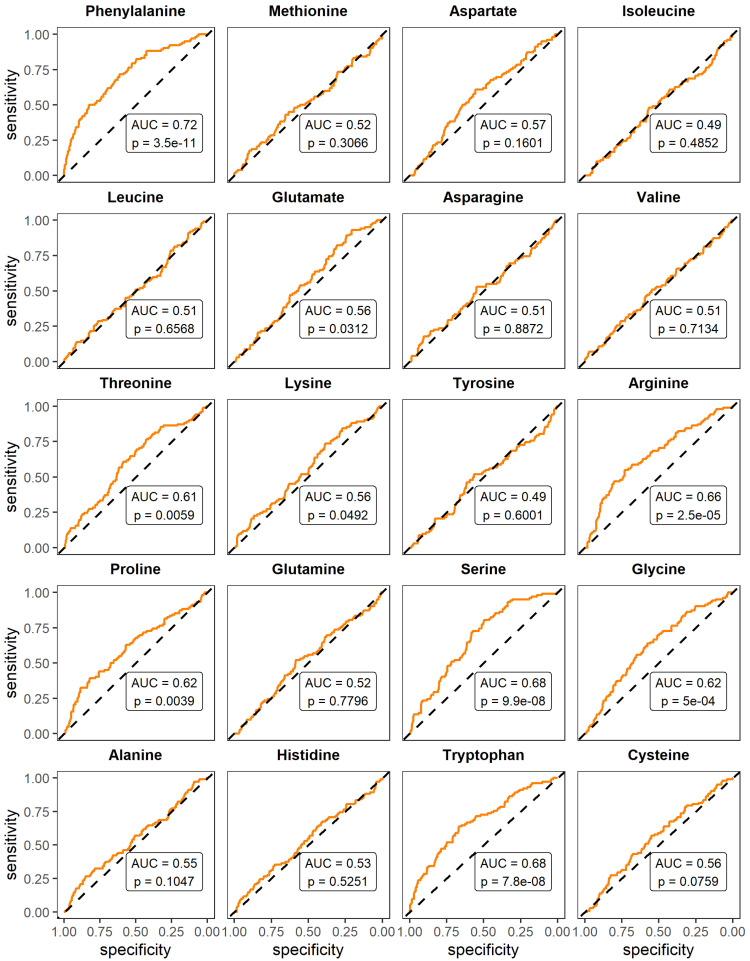
Ability of circulating-amino-acid levels to predict adverse outcomes (intensive-care-unit admission or hospital death). Univariate logistic regressions were used, and areas under the receiving-operator-characteristic curve (ROC_AUC) were computed. A ROC_AUC close to 0.5 means that the predictor is ineffectual, a ROC_AUC over 0.8 means the predictor is good, and a ROC_AUC of 1 means a perfect predictor. Blood was drawn on average 19 days before death and 3 days before ICU admission.

**Table 1 metabolites-13-00201-t001:** Sample characteristics by COVID-19 severity.

Variables (Unit)	*n*=	Mild (*n* = 159)	Moderate (*n* = 353)		Severe (*n* = 224)		*p*-Value
Age (years)	736	55.7 (44.2–67.2)	67.3 (51.8–81.9)	#	66.0 (54.8–73.6)	#	2.4 × 10^−10^
Sex (female)	736	86 (54%)86 (54%)	168 (48%)		66 (29%)	#&	1.0 × 10^−6^
BMI (kg/m^2^)	372	28.1 (24.7–31.7)	26.7 (23.6–30.4)		28.7 (25.6–33.9)	&	0.0183
Vaccinated (yes)	321	91 (90%)	104 (84%)		83 (86%)		0.3938
Number of vaccine doses (%, 1/2/3 doses)	277	23%/58%/19%	41%/58%/1%	#	37%/61%/1%	#	1.5 × 10^−6^
Outpatient (yes)	736	80 (50%)	3 (1%)	#	1 (0%)	#	1.2 × 10^−66^
CRP (mg/L) *	391	16.4 (1.9–56.2)	42.3 (18.5–89.1)	#	96.1 (45.4–173.2)	#&	1.2 × 10^−12^
ALT (U/L)	391	25.5 (18.8–48.2)	27.0 (17.0–51.0)		35.5 (22.2–69.0)	&	0.0061
Glucose (mmol/L)	451	5.65 (5.23–6.57)	5.90 (5.10–7.68)		7.40 (6.10–9.90)	#&	2.9 × 10^−9^
Creatinine (mmol/L)	519	72.0 (60.0–87.5)	74.0 (57.0–94.0)		84.0 (63.5–125.0)	#&	0.0011
Haemoglobin (g/L)	508	137 (123–151)	125 (112–137)	#	120 (103–133)	#&	1.6 × 10^−7^
Urea (mmol/L)	487	4.40 (3.00–5.50)	5.80 (3.80–9.05)	#	8.00 (5.47–14.72)	#&	5.3 × 10^−13^
Albumin (g/L)	439	41.0 (39.0–44.0)	34.0 (31.0–37.0)	#	30.0 (27.0–34.0)	#&	2.7 × 10^−27^
Bilirubin (μmol/L)	380	7.00 (5.00–9.00)	7.00 (5.00–10.00)		8.00 (6.00–11.00)		0.0600
WBC (×10^9^/L)	507	6.00 (4.60–7.32)	6.50 (5.10–8.60)		8.55 (6.15–11.72)	#&	1.6 × 10^−10^
LDH (U/L)	291	251 (212–324)	303 (243–383)	#	386 (343–585)	#&	2.8 × 10^−10^
Procalcitonin (ug/L)	168	0.07 (0.05–0.10)	0.11 (0.08–0.17)	#	0.23 (0.16–0.67)	#&	1.4 × 10^−10^
D-Dimer (ug/L)	160	637 (447–860)	856 (543–1213)	#	1280 (817–2181)	#&	1.2 × 10^−5^
Temperature (C)	543	37.0 (36.8–37.5)	37.0 (36.7–37.8)		37.4 (36.7–38.5)		0.0281
SBP (mmHg)	576	130 (119–147)	128 (115–144)		125 (110–138)	#	0.0199
DBP (mmHg)	576	81.0 (70.0–87.2)	76.0 (67.0–84.0)	#	72.0 (66.0–80.0)	#&	2.7 × 10^−5^
Heart rate (beats/min)	582	97.0 (84.0–105.2)	92.0 (78.0–107.0)		96.5 (82.0–109.0)		0.1000
SaO_2_ (%)	531	97.0 (95.0–99.0)	95.0 (93.0–97.0)	#	93.0 (89.0–95.0)	#&	6.9 × 10^−23^
Oxygen (yes)	565	0 (0%)	124 (39%)	#	132 (71%)	#&	1.6 × 10^−24^

Data are presented as median (Q1-Q3) for continuous variables and number (%) for dichotomous variables. Characteristics were compared across severity groups using Krustal–Wallis test for continuous variables and chi-square test otherwise. Post hoc comparisons were made using the same tests but using a Bonferonni-corrected *p*-value threshold of *p* < (0.05/3); &: significant vs. moderate, #: significant vs. mild. BMI: body mass index, CRP: C-reactive protein, ALT: alanine aminotransferase, WBC: white blood cells, SBP: systolic blood pressure, DBP: diastolic blood pressure, LDH: lactate dehydrogenase, SaO_2_: arterial oxygen saturation. * For CRP, patients receiving tomizumab were excluded because it negates the measurement.

**Table 2 metabolites-13-00201-t002:** Univariate association between circulating-amino-acid levels and the odds of presenting mild vs. severe and moderate COVID-19 symptoms.

Amino Acid	Mild vs. Severe			Mild vs. Moderate		
	OR	95%CI	*p*-Value	OR	95%CI	*p*-Value
Phenylalanine	4.14	(2.79–6.13)	1.5 × 10^−12^	1.68	(1.33–2.11)	1.0 × 10^−5^
Methionine	2.72	(1.96–3.77)	2.2 × 10^−9^	1.67	(1.34–2.09)	4.3 × 10^−6^
Aspartate	1.82	(1.44–2.30)	7.3 × 10^−7^	1.14	(0.94–1.39)	0.1745
Isoleucine	1.64	(1.31–2.05)	1.8 × 10^−5^	1.18	(0.97–1.43)	0.0997
Leucine	1.57	(1.26–1.97)	7.0 × 10^−5^	1.15	(0.95–1.39)	0.1508
Glutamate	1.51	(1.19–1.90)	0.0006	1.18	(0.97–1.44)	0.1055
Asparagine	1.37	(1.09–1.71)	0.0065	1.17	(0.97–1.42)	0.1039
Valine	1.35	(1.09–1.67)	0.0060	1.08	(0.89–1.31)	0.4255
Threonine	1.18	(0.96–1.46)	0.1184	1.19	(0.98–1.46)	0.0814
Lysine	1.17	(0.95–1.44)	0.1427	1.09	(0.90–1.32)	0.3638
Tyrosine	1.11	(0.90–1.36)	0.3288	0.92	(0.76–1.11)	0.3813
Arginine	0.96	(0.78–1.17)	0.6763	0.97	(0.80–1.16)	0.7171
Proline	0.75	(0.61–0.93)	0.0078	0.73	(0.60–0.88)	0.0010
Glutamine	0.68	(0.55–0.85)	0.0005	0.89	(0.74–1.08)	0.2427
Serine	0.68	(0.55–0.84)	0.0003	0.86	(0.72–1.04)	0.1237
Glycine	0.65	(0.53–0.81)	9.5 × 10^−5^	0.89	(0.74–1.08)	0.2362
Alanine	0.57	(0.46–0.72)	1.1 × 10^−6^	0.68	(0.57–0.83)	9.7 × 10^−5^
Histidine	0.45	(0.35–0.57)	1.7 × 10^−10^	0.50	(0.40–0.62)	9.1 × 10^−11^
Tryptophan	0.38	(0.29–0.50)	9.7 × 10^−13^	0.57	(0.47–0.70)	7.2 × 10^−8^
Cysteine	0.34	(0.26–0.45)	7.2 × 10^−15^	0.45	(0.37–0.56)	4.6 × 10^−13^

Results are presented as odds ratios per 1 standard-deviation (SD) increase in amino-acid concentration.

**Table 3 metabolites-13-00201-t003:** Results from the multivariate logistic-regression model to predict adverse outcomes.

Term	Estimate	Std. Error	Statistic	df	*p* Value
(Intercept)	8.54	3.93	2.17	355.57	0.0306
CRP	0.01	0.00	3.58	201.67	0.0004
Urea	0.10	0.03	3.29	332.92	0.0011
SaO_2_	−0.10	0.04	−2.73	339.54	0.0068
SBP	−0.02	0.01	−2.29	417.97	0.0228
Phenylalanine	2.41	0.62	3.89	410.55	0.0001
Tryptophan	−1.91	0.77	−2.48	401.59	0.0136
Arginine	−1.68	0.59	−2.86	450.63	0.0044
Lysine	2.83	0.94	3.00	437.35	0.0028
Glutamate	−0.65	0.37	−1.73	436.87	0.0845
Serine	−1.59	0.93	−1.71	441.13	0.0881

Variables were scaled and imputed using multivariate imputation (predictive mean matching for continuous variables and logistic regression for ordinal variables). Variables with ≥50% missing values were dropped. The variables to include in the final model were selected from forward and backward stepwise-regression run on the 50 multiple-imputation datasets. CRP: C-reactive protein, SaO_2_: arterial oxygen saturation, SBP: systolic blood pressure.

## Data Availability

Restrictions apply to the availability of these data. Data was obtained from the Biobanque Québécoise COVID-19. Access to the data can be requested through their website: https://www.bqc19.ca/en accessed on 28 September 2022.

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
