# Peer review of "Association between Circulating Amino Acids and COVID-19 Severity"

_metabolites, 2023, doi:10.3390/metabo13020201_

Round 1
Reviewer 1 Report
Major concerns:
- The plasma samples were subjected to a comprehensive data acquisition that detected 1438 metabolites. However, only amino acids, TCA, and urea cycle-related metabolites were used in this study. This is quite unfortunate because many potential pathways and metabolites associated with the severity of COVID-19 were not investigated.
- In Figure 4, a series of univariate logistic regression was used to compute AUC. After finding potential amino acids that may be associated with the severity of COVID-19, why not make a (multivariate) model (with proper cross-validation) capable of accurately classifying severity groups?
- It is inconsistent regarding the purposes/sources when "Table 1" is cited in the main text (e.g., lines 41, 187, 260). It looks pretty confusing to me. Besides, the table "Table 1. Summary of the associations between circulating amino acid concentrations and COVID- 49 19 symptom severity reported in the literature" is currently placed in the Introduction. I wonder if it is a common practice. I would, instead, add it into part of the method and material for reasoning why amino acids were chosen to be studied or discussion to clarify the roles of amino acids in the literature.
Minor comments:
- The scale of the OR and SD in some panels of Figure 1 should be re-adjusted to present the data better. It is especially true for the comparison between mild and moderate groups. For example, we cannot clearly see or approximate the OR of phenylalanine in the "mild vs. moderate" panel.
- Please add Figure legends for supplementary figures 1 and 2. Besides, it is better to use the journal template for supplementary figures and tables.
Other suggestion:
The current statistical analysis method is more suitable for getting better insights into the association among severity groups. It is also interesting to see that the association of amino acids and severity is not significantly influenced by other clinical and treatment factors. However, the data set is relatively large (736 patients), and the number of covariates is redundant. I believe an investigation based on proper modeling (including clinical covariate and circulating metabolites) for severity classification/prediction is also possible. Besides, it appears that a non-linear model would be more suitable for this task. Please consider this suggestion.
Author Response
Major concerns:
- The plasma samples were subjected to a comprehensive data acquisition that detected 1438 metabolites. However, only amino acids, TCA, and urea cycle-related metabolites were used in this study. This is quite unfortunate because many potential pathways and metabolites associated with the severity of COVID-19 were not investigated.
In the past few years, our research group has been interested in the association between circulating amino acids and visceral adiposity as well as metabolic dysfunction. In the context of recent evidence showing that obesity, and particularly visceral adiposity, is associated with severe COVID-19 symptoms, we aimed to investigate whether circulating amino acids, viewed here as metabolic surrogates of abdominal obesity, were associated with COVID-19 severity. This is the research question that was approved for funding by the peer-review committee of our hospital foundation.
We agree that the comprehensive metabolomics dataset available from the Biobanque Québécoise COVID-19 (BQC19) is very rich and could be use for untargeted analyses to answer broader, highly interesting questions. The data are available upon request for researchers directly through the BQC19 infrastructure.
Other studies have used untargeted metabolomics to identify metabolic signatures related to the severity of COVID-19. Papers reviewing this literature are now available (e.g. Arjmand et al. 2021, Hasan et al. 2021, Lin et al. 2021 and Mussap et al. 2021). One group of metabolites that almost systematically stands out as being altered in severe vs mild cases are amino acids. Of note, important discrepancies are noted in the results for each individual amino acid, which reinforces the need for a study targeting these molecules. In this context, we respectfully suggest that it is of interest to summarize existing data on amino acids and covid severity and analyze this specific group of metabolites in the largest dataset available so far.
To better explain the rationale of the study and our decision to investigate amino acids and related metabolites, we added the following to the methods (page 3):
“We focus on circulating amino acids for two reasons. First, abdominal obesity has previously been associated with COVID-19 severity, and we have previously demonstrated that circulating concentrations of some amino acids are correlated with central fat accumulation. Second, existing literature shows that amino acids, as a group, are almost invariably altered in the context of severe COVID-19, although results for specific amino acids are heterogeneous.”
- In Figure 4, a series of univariate logistic regression was used to compute AUC. After finding potential amino acids that may be associated with the severity of COVID-19, why not make a (multivariate) model (with proper cross-validation) capable of accurately classifying severity groups?
We agree that a multivariate logistic model would be interesting in the context of our study. Therefore, we built such a model to predict adverse outcome using clinical characteristics and circulating amino acids that were significant in univariate models. The methods and the results from this analysis can now respectively be found on pages 4 and 12 of the manuscript.
- It is inconsistent regarding the purposes/sources when "Table 1" is cited in the main text (e.g., lines 41, 187, 260). It looks pretty confusing to me. Besides, the table "Table 1. Summary of the associations between circulating amino acid concentrations and COVID- 49 19 symptom severity reported in the literature" is currently placed in the Introduction. I wonder if it is a common practice. I would, instead, add it into part of the method and material for reasoning why amino acids were chosen to be studied or discussion to clarify the roles of amino acids in the literature.
The summary Table was removed from the introduction and placed in supplementary material. It is no longer cited in the introduction, but we still refer to it in the discussion to position our results regarding existing literature.
Minor comments:
- The scale of the OR and SD in some panels of Figure 1 should be re-adjusted to present the data better. It is especially true for the comparison between mild and moderate groups. For example, we cannot clearly see or approximate the OR of phenylalanine in the "mild vs. moderate" panel.
We agree the scales for Figure 1 could make it hard to see the approximate OR for some amino acids. To address this, we changed the units for the x-axis in Figure 1. We decided to keep the same x-axis scale for a given amino acid for both the severe vs mild and moderate vs mild comparison. This allows the reader to visually assess the extent of the difference in OR between the 2 comparisons for each amino acids and to see the effect of adjustments for clinical variables. To get precise numbers, the univariate OR are presented explicitly in Table 2.
- Please add Figure legends for supplementary figures 1 and 2. Besides, it is better to use the journal template for supplementary figures and tables.
We added the legends for the supplementary figures.
Other suggestion:
The current statistical analysis method is more suitable for getting better insights into the association among severity groups. It is also interesting to see that the association of amino acids and severity is not significantly influenced by other clinical and treatment factors. However, the data set is relatively large (736 patients), and the number of covariates is redundant. I believe an investigation based on proper modeling (including clinical covariate and circulating metabolites) for severity classification/prediction is also possible. Besides, it appears that a non-linear model would be more suitable for this task. Please consider this suggestion.
We agree that multivariate models can bring interesting insight and that our dataset is large enough to make this type of analysis possible. To address this, we computed a multivariate logistical model to predict adverse outcomes. The methods and the results from this analysis can now be found on pages 4 and 12 of the manuscript respectively.
Reviewer 2 Report
The authors tried to evaluate the relation of circulating amino acids to COVID-19 severity. They found that COVID-19 severity would alter the amino acids profiles in spite of lacking prognostic impact. Although this is a negative result paper, the article still provides the new information about the changes of amino acids profiles in patients with severe COVID-19 symptoms.
Author Response
The authors tried to evaluate the relation of circulating amino acids to COVID-19 severity. They found that COVID-19 severity would alter the amino acids profiles in spite of lacking prognostic impact. Although this is a negative result paper, the article still provides the new information about the changes of amino acids profiles in patients with severe COVID-19 symptoms.
We thank Reviewer #2 for these comments and for appreciating the novelty of our work.
Reviewer 3 Report
The manuscript by Maltais-Payette I et al., entitled "Association between circulating amino acids and COVID-19 severity" provides significant data in the context of COVID-19. However there are some omissions that should be addressed.
1. Considering that there are different types of obesity (Mayoral LP, doi: 10.4103/ijmr.IJMR_1768_17), please specify the type of obesity that the patients had, and comment this, in discussion.
2. The authors must obtain the predicting visceral fat in patients studied and relate it to their results.
3. Considering that the areas under the receiving operator characteristic curve (ROC_AUC) change as a function of time during the course of the disease (Geraili Z, et al (2022) doi: 10.1177/03000605221102217), please specify the time at which samples were obtained to predict adverse outcomes in circulating amino acids (figure 4), and comment this in the discussion.
4. What are the cutoff points? in Supplemental Table 3: Ability of baseline characteristics to predict adverse outcomes.
5. The authors must indicate the limitations of their work.
Author Response
The manuscript by Maltais-Payette I et al., entitled "Association between circulating amino acids and COVID-19 severity" provides significant data in the context of COVID-19. However, there are some omissions that should be addressed.
- Considering that there are different types of obesity (Mayoral LP, doi: 10.4103/ijmr.IJMR_1768_17), please specify the type of obesity that the patients had, and comment this, in discussion.
We agree that obesity is associated with a wide variability of metabolic health parameters and that it is of interest to decipher metabolically healthy from metabolically unhealthy obese participants. Unfortunately, we have limited information on the metabolic health of the patients in our cohort. For instance, we do not have access to variables describing body fat distribution (i.e. waist circumference or visceral adipose tissue area).
The following analyses were performed to address this suggestion. To estimate whether participants living with obesity were metabolically unhealthy obese (MUO) or metabolically healthy obese (MHO), we used the following metabolic comorbidities: diabetes, arterial hypertension and coronary arterial disease. Patients with a diagnosis of obesity (or a measured BMI ≥ 30 kg/m2) and at least 1 metabolic comorbidity were considered as MUO, whereas those with only obesity (or a BMI ≥ 30 kg/m2) were classified as MHO.
We discussed the prevalence of both obesity types across disease severity in the result section (page 5) and we acknowledge the lack of body fat distribution measurement in the discussion as a limitation.
- The authors must obtain the predicting visceral fat in patients studied and relate it to their results.
Unfortunately, we do not have the variables necessary to assess or predict the visceral fat of our patients. This is a limitation of our work and, therefore, acknowledge this in the limitation section of the paper.
- Considering that the areas under the receiving operator characteristic curve (ROC_AUC) change as a function of time during the course of the disease (Geraili Z, et al (2022) doi: 10.1177/03000605221102217), please specify the time at which samples were obtained to predict adverse outcomes in circulating amino acids (figure 4), and comment this in the discussion.
This information was added in the legend of Figure 4 and commented in the discussion.
- What are the cutoff points? in Supplemental Table 3: Ability of baseline characteristics to predict adverse outcomes.
The predicting ability of baseline characteristics in Supplemental Table 3 (and that of amino acids in Figure 4) is reported as the area under the receiver operating characteristic (ROC) curve. Such curves are drawn from the sensitivity and specificity to distinguish groups established with every possible threshold in the continuous variable of interest. A threshold in the continuous variable is sometimes chosen based on clinical expertise or a statistical method (such as the Youden’s index), but in our study, we decided to not select a threshold. This is because the amino acid measurements are in relative (and not absolute) concentrations. This means that optimal thresholds in amino acid concentration would not be expected to be similar or comparable between metabolomic studies using different methodological approaches. Since the prediction ability of clinical variables was computed to compare it to that of amino acids, we also reported it as the ROC_AUC, without any threshold.
To address this comment, we now expand our explanation of the ROC_AUC in the methods section and we justify why we did not determine optimal thresholds (page 4).
- The authors must indicate the limitations of their work.
A limitation section can be found in the discussion of the paper.
Round 2
Reviewer 1 Report
I think the manuscript has been significantly improved thanks to additional analyses and data interpretation.
It is, however, still biased toward the amino acids and the explanations are not convincing given the richness and depth of the data available to the authors.
Besides, the putative functional interpretation is not scientifically sound and contains potentially misleading information (Fig. 3), given that much data were kept hidden.
It's worth pointing out that amino acids are easily detected in any conventional LC-MS analyses. This is a well-known technical "bias," which leads to many publications claiming amino acids are "biomarkers" of virtually hundreds, if not thousands, of diseases. However, their level differences are complex to interpret accurately due to many biological routes of formation and deletion through metabolism and catabolism.
Finally, I believe that the abstract can be improved, e.g., by reducing "introductory content" and increasing the findings from the newly conducted multivariate analysis.
Author Response
Reviewer’s comment:
I think the manuscript has been significantly improved thanks to additional analyses and data interpretation.
It is, however, still biased toward the amino acids and the explanations are not convincing given the richness and depth of the data available to the authors.
Besides, the putative functional interpretation is not scientifically sound and contains potentially misleading information (Fig. 3), given that much data were kept hidden.
Response:
We thank review 1 for the review of our manuscript.
This manuscript focuses only on amino acids and does not aim to find the metabolite(s) most strongly associated with COVID-19 severity. Instead, we aim to determine which amino acid circulating concentrations are associated with COVID-19 symptom severity and to determine whether they could be relevant predictors of adverse outcomes. The aim of this paper stems from previous work linking both circulating amino acids and COVID-19 severity to visceral adiposity, which is the primary focus of our research group. An untargeted metabolomics analysis would move us away from this perspective.
We recognise that an untargeted metabolomics analysis would be valuable, but it would answer a different research question than ours and it would warrant a completely new manuscript. We respectfully request to keep the focus of the analysis as is. However, we have made many changes to the manuscript in order to better covey our intent and to acknowledge the caveats associated with this approach.
- To clarify our perspective, we added the following sentence to the introduction: “This study is targeted on amino acids and does not aim to find the metabolite (or group of metabolites) most strongly associated with COVID-19 severity.”
- We removed Figure 3 (and related text) from the manuscript because we agree that it could lead the reader to think that the pathways presented are more strongly associated with COVID-19 severity than other pathways not investigated, or that they are causally responsible for the alterations in circulating amino acids.
Reviewer’s comment:
It's worth pointing out that amino acids are easily detected in any conventional LC-MS analyses. This is a well-known technical "bias," which leads to many publications claiming amino acids are "biomarkers" of virtually hundreds, if not thousands, of diseases. However, their level differences are complex to interpret accurately due to many biological routes of formation and deletion through metabolism and catabolism.
Response:
We acknowledge that most amino acids are well detected by LC-MS analyse and that their ease of detection contributed to the vast literature around this group of metabolites. We are not suggesting that amino acids are more strongly associated with COVID-19 severity or better predictors of adverse outcomes than other metabolites. To clarify this, we integrated the reviewer’s comment to the discussion section as follows:
“Finally, we acknowledge that amino acids are more easily and frequently measured in metabolomics studies compared to other classes of metabolites and this probably contributed to the abundance of studies linking amino acids and COVID-19. Whether amino acids are better than other metabolites at predicting COVID-19 severity or adverse outcomes should be investigated further.”
Reviewer’s comment:
Finally, I believe that the abstract can be improved, e.g., by reducing "introductory content" and increasing the findings from the newly conducted multivariate analysis.
Response:
We added results from the multivariate analysis to the abstract.
Reviewer 3 Report
P7 L207.- Correct the caption of figure 2 and its typographical errors.
Table 1 and 4, P10 L271, ST2.- correct "saturation" by arterial oxygen saturation (%) (SaO2) where appropriate.
Authors should improve the abstract with data from the multivariate logistic regression model.
Author Response
Reviewer's comment:
P7 L207.- Correct the caption of figure 2 and its typographical errors.
Table 1 and 4, P10 L271, ST2.- correct "saturation" by arterial oxygen saturation (%) (SaO2) where appropriate.
Authors should improve the abstract with data from the multivariate logistic regression model.
Response:
We thank reviewer 3 for a thorough review of the manuscript. Changes were done as requested.
Round 3
Reviewer 1 Report
I still cannot fully agree with the scientific reasoning of the authors regarding the rationale behind this project. However, I respect the potential merit of this work to the broad scientific community. Besides, misleading data interpretation and information have been removed, and potential issues of the content have been clarified, which makes it suitable to be accepted.